# Evaluating the impact of climate communication activities by scientists: What is known and necessary?

Frances Wijnen[1], Madelijn Strick[2], Mark Bos[1], Erik van Sebille[1]

[1]Freudenthal Institute, Utrecht University, Utrecht, 3584 CC, Netherlands
[2]Faculty of Social and Behavioural Sciences, Utrecht University, Utrecht, 3584 CC, Netherlands

*Correspondence to*: Erik van Sebille (E.vanSebille@uu.nl)

**Abstract.** Climate scientists and others are urged to communicate climate science in a way that non-scientific audiences can understand, that makes it more relevant to their lives and experiences, and that inspires them to act. To achieve this, climate scientists undertake a range of climate communication activities to engage people with climate change. With the effort and time spent on climate communication activities, comes the need to evaluate the outcomes, impact and effectiveness of such efforts. Here, we aimed to gain insight into the impact and effectiveness of climate communication efforts by scientists by conducting a systematic literature review. However, our most important finding is that there are hardly any studies in which climate communication activities by scientists are evaluated: we found only seven articles over the past ten years. We analyze these articles for the role of the scientists, the audiences reached and the reported outcomes and impact of the activities. We end our study with several recommendations that should be considered when setting up studies on evaluating the impact of climate communication activities by scientists.

## 1 Introduction

Climate change is one of today's greatest challenges that people around the world face (IPCC, 2022; Schneider, 2011). The consequences of climate change, such as extreme weather events, sea level rise, and impacts on ecosystems and biodiversity are expected to (further) increase in the coming years (IPCC, 2022). The Intergovernmental Panel on Climate Change (IPCC) emphasizes the need for rapid, far-reaching and unprecedented actions to limit the rise of temperatures above 1.5ºC compared to pre-industrial levels (IPCC, 2018).

For people to take such actions, it is important that they are aware that climate change is happening, that its causes are anthropogenic, and what adaptation and mitigation actions are needed to address climate change (Hassol, 2008). Therefore, climate scientists and others are urged to communicate climate science in a way that non-scientific audiences can understand, that makes it more relevant to their lives and experiences, and that inspires them to act (Corner & Clarke, 2017; Corner et al., 2018; Kumpu, 2022; Dechezleprêtre et al., 2022).

To achieve this, climate communication activities are needed that engage people with climate change (Kumpu, 2022). Engagement with climate change may be defined broadly as individuals' evaluation of and response to climate change, which comprises cognitive, emotional, and behavioral components. That is, engagement involves what people think, feel, and do about climate change (Lorenzoni et al., 2007 in Whitmarsh et al., 2013, p. 4). In this study we use a broad description of engagement with climate change, which includes psychological factors that might impact what people think, feel, and do

about climate change. These factors include for example attitude towards climate change, motivation to act, perceived capability in taking action (often referred to as self-efficacy based on Bandura's concept) and social norm (see Van der Linden et al., 2015 for a description of several of these factors).

Climate communication activities to engage people with climate change are very diverse. Examples include exhibitions on sustainable eating during festivals (Kluczkovski et al., 2020), informing the public about climate change through media, such

as TV broadcasts (Calyx & Low, 2020), and using participatory arts (Burke et al., 2018). With the effort and time spent on this diversity of climate communication activities comes the need to evaluate the impact and effectiveness of such efforts (Grand & Sardo, 2017). However, to our knowledge, there is no comprehensive overview that describes this impact and effectiveness.

## 1.1 Aim of this study

We conducted a literature review where we aimed to answer the following research question: What is known about the *impact* of climate communication activities *by scientists* on people's engagement with climate change? While there are other actors who engage in climate communication (e.g. communication professionals, knowledge brokers), we chose to specifically focus on climate communication activities *by scientists* because research shows that scientists are seen by the public as trusted information producers (Dziminska et al., 2021) and the public believes that scientists should increase their

communication efforts (Cologna et al., 2021). On the other hand, scientists are often hesitant to engage in climate communication activities because they feel not sufficiently trained (Rozance et al 2020), or for fear of hurting their credibility (Kotcher et al., 2017) and potentially being accused of "advocacy".

Cologna et al. (2021) describes arguments in favor of and against advocacy by scientists. Arguments against advocacy include that advocacy would undermine the credibility of scientists because it contradicts the scientific ideal of neutrality

(Lackey, 2007; Nielsen, 2001) and that it negatively influences scientists' ability to conduct science (e.g., due to time-constraints; Nelson & Vucetich 2009). Arguments in favor of advocacy include that it is appropriate when no advocacy could be harmful to society (Douglas, 2009), that scientists as citizens have the responsibility to engage in political and public debates (Lubchenco, 1998, 2017), and that science is never value-free. Therefore, science and advocacy are impossible to differentiate (Elliott & Resnik, 2014, Schmidt, 2015). It is important to carefully consider these different

arguments, given the need for strong climate action while maintaining trust and credibility in science and scientists. Insight into the impact of climate communication activities could inform scientists about when and in what way it is effective and appropriate to partake in such activities.

## 2 Theoretical Framework

### 2.1 Importance of evaluation

Grand and Sardo (2017) describe that it is important to assess the effectiveness and impact of science communication activities. However, high-quality evaluation can do more than assessing. It can promote innovation, change, and be used to critically reflect on the process of engaging the public (Wilkinson & Weitkamp, 2016). Or as Jensen (2015) describes it, evaluation can help scientists understand which aspects of science communication are working, in what way, for which audiences and why. However, evaluating science communication efforts is not easy. Good evaluation means that clear

objectives and appropriate evaluation methods are required. This means that evaluating impact goes beyond counting the number of people involved, or impressions based on informal chats (Grand & Sardo, 2017).

Peeters et al., (2022) distinguish three levels of evaluation. The first level, *output,* focuses on evaluating material results such as the number of people attending an event, sales numbers, or where participants are coming from. The second level, *outcomes,* focuses on the direct effects of a communication activity on the public, such as whether the public learned

something, was inspired, or is motivated to act. The third level, *impact*, focuses on evaluating the impact of communication activities on society or the effect on a specific target group over a longer period. Evaluating impact can focus on (changes in) societal norms, values, or actions. According to Peeters et al., (2022), evaluating impact could be achieved by evaluating outcomes over a longer period and contextualizing the results in a societal frame. We use these three levels of evaluation to review the climate communication activities that are described in the literature.

### 2.2 Role of scientists in climate communication

A strong science-policy interface is required to achieve the global climate targets. However, it is yet unclear if and to what extent (climate) scientists are willing and required to engage in climate communication activities. In addition, scientists can take on different roles in climate communication activities. In science communication three models are often used to describe the interaction between scientists and the public (see Metcalfe, 2019). The deficit model represents a one-way form

of communication where scientists inform the public about science and scientific findings. In this model, the communicator/scientist takes on a role of expert or information source, where the most important goal is to inform the public about science. This may sometimes be an appropriate role, for example in disaster risk communication.

The dialogue model describes science communication, not as a one-way approach but, as a dialogue between scientists and the public. This model is characterized by three main features: (1) scientists are willing to engage in a dialogue with the

public to help explain the science and its meaning (Wynne, 2006), (2) scientists are willing to consult the public and are open to responses and feedback on their concerns, perceptions, and questions about science, and (3) scientists acknowledge that 'the public' may have useful knowledge and ideas that can help in scientific research. In the dialogue model, scientists still take on the role of expert but are now interested and willing to hear about questions and perceptions from the public to get a

better understanding of the societal response. This is an appropriate role when for example designing effective risk communication strategies with communities and could also help the scientists to improve their research (Boon et al., 2022). The third model is the participatory model, which is similar to the dialogue model. However, the participatory model recognizes and acknowledges different publics as equal with scientists and policymakers. The participatory model describes a clearer shift in power from the scientists to the public. In this model, the scientist is no longer 'the expert' but someone with an interest in a scientific topic who likes to engage in discussion with others (non-scientists) about this topic. Again, we used these three roles of scientists to review the climate communication activities that are described in the literature.

## 3 Method

### 3.1 Literature Search

To identify relevant literature for answering our research question, we conducted a literature search based on title, abstract and keywords with synonyms for the terms: climate communication, climate change, and engagement. For a full list of the used keywords, see Table 1. The databases that we used were Web of Science, Scopus and PsycInfo. We focused our search on literature of the past ten years (2012-2022) because we expect that this reflects current developments in research into climate change communication. We selected peer-reviewed articles that were written in English. We imported the found literature into the Mendeley reference manager program. After removing duplicates, a set of 819 documents remained.

| Search term | Used keywords |
| --- | --- |
| Climate Communication | "Scien* communicat*" OR "environmental communicat*" OR "citizen engagement" OR "climate communicat*" OR "public involvement" OR "outreach" OR "Public engagement" OR (public "NEAR/4" communicat*) |
| Climate change | "Climate change" OR "global warming" OR "global heating" OR "climate crisis" OR "climate emergency" OR "climate challenge" OR "climate science" |
| Engagement | attitude OR perception OR belief OR opinion OR affect OR emotion OR "social norm" OR "subjective norm" OR anxiety OR fear OR concern OR enjoyment OR "self-efficacy" OR "perceived capability" OR hope* OR cognit* OR value OR knowledge OR comprehen* OR intent* OR motivation OR behaviour OR behavior OR "scientific literacy" OR "scientific skill*" |

**Table 1: Search terms used.**

Next, we reviewed the documents based on the title and abstract. We used two selection criteria: (1) the study focuses on climate communication activities *by scientists*, (2) the study describes *the output, outcome, and/or impact* of climate communication activities. The measurements used to evaluate the output, outcome or impact could be either quantitative or qualitative. If these two criteria were not met, the document would be discarded. In some cases, it was difficult to determine, based on the title and abstract, whether the climate communication activity involved scientists. In these cases, we decided to

include or exclude based on the full text at a later stage. The analysis based on the title and abstract resulted in a remaining set of 66 documents. In four cases, we had no access to the full text. Therefore, we evaluated 62 full texts for our analysis.

## 3.2 Inclusion of documents

The 62 documents were reviewed by the first author to determine whether they fit the inclusion criteria. After evaluating 35 documents, the third author evaluated 10 documents that were already reviewed by the first author to see whether he came to

a similar conclusion about the inclusion of documents. These results matched the conclusions of the first author. Therefore, the first author finished reviewing the remaining documents. After analyzing all documents, a few remaining documents for which there were doubts about inclusion, were discussed in the research team and a decision for inclusion was made. These steps led to the exclusion of 55 documents. The most important reasons for exclusion were:

1. The climate communication activities did not involve *scientists* (e.g., Geiger et al., 2017; Van Swol et al., 2019).

2. There is no e*valuation of the impact* of climate communication activities (e.g., Oosterman, 2016; Cologna et al., 2021).

## 3.3 Analysis

The steps described above lead to the inclusion of seven documents (Calyx & Low, 2020; Illingworth & Jack, 2018; Jacobson et al., 2016; Kluczkovski et al., 2020; Luís et al., 2018; Pathak et al., 2021; Peltola et al., 2020). Although we

initially aimed to evaluate the impact of climate communication activities by scientists, the final inclusion of seven documents limits our possibilities for analysis. However, we do think that our finding that there is very little research on the impact of climate communication activities by scientists is important. This suggests that evaluating climate communication activities by scientists has over the past 10 years not been a focus of scientific research.

For our analysis we describe for each included study: (1) the role of the scientist in the climate communication activity, (2)

the reached audiences, and (3) outcome variables that were evaluated and conclusions regarding impact. A description of the included studies can be found in Table 2.

| # | Goal of the study | Climate Communication activity | Role of the scientist | Target audience |
|---|---|---|---|---|
| 1 | The goal of this case study was to explore the efficacy of an interdisciplinary learning experience integrating science and art students to enhance the curriculum of climate change and potentially other sustainability challenges. | The students participated in an orientation and a one-day field trip to the Marine Lab that included group discussions, lectures by scientists, and an artist-led art-making project creating found- object collages to represent climate change processes. Students subsequently worked on small group activities to develop communication material for public visitors to the Marine Lab | Expert: Two artists and two biological scientists - Both science and art instructors walked across Seahorse Key with the students. They answered questions about coastal ecology and climate change, as well as about collage making and creative idea finding. | Nine students from an advanced fine arts class and nine from a natural resource management class. |
| 2 | This study seeks to develop a framework through which experts (e.g., scientists) can engage in a dialogue with underserved audiences about environmental change. It uses poetry to help establish this framework, and presents an interpretation of how we can use this poetry to better understand the audiences, and how they perceive environmental change | Workshops (three sessions at two places) were participants created poetry on climate change | Dialogue: Scientists ranging from early career scientists to professors. They created poetry together with the target group about climate change and joined in the discussions that came up during the sessions. | People from underserved communities. Multiple workshops were organized at two places. At the first location, the total number of participants was 11. At the second location, it varied from 10 to 20 |

| | | | |
|---|---|---|---|
| 3 | The aim of the 'Take a Bite' exhibit was to engage with the public to raise awareness about the impact of food choices on the climate, promote sustainable food consumption behaviors, and empower consumers with accessible knowledge to make informed decisions, as well as increasing consumer acceptance of interventions to help reduce food greenhouse gas emissions. In addition, the 'Take a Bite' exhibit was developed as an opportunity for individuals to engage with researchers while conveying the message that individual choices can make a difference to tackle climate change | An interactive exhibition on the potential opportunities for lowering greenhouse gas emissions of food production. | Expert: Scientists from relevant disciplines worked together to design and build the exhibit. Expert communicators (not the scientists) were present during the exhibition to talk to visitors | Teachers, students, families and members of the public who visited the two events where the exhibition took place. These events were a summer science exhibition and a music festival. The 'Take a Bite' stand engaged with approximately 6868 people, 64% adults (age: 20+), 16.8% teenagers (aged 13–19) and 18% children (age: <13) |
| 4 | This essay draws on the perspective of participants, speakers and organizers of 17 in-person outreach events conducted across India in 2018 and 2020. The goal is to share insights and recommendations for future IPCC events in India and other developing country contexts | Differs per event, lectures, workshops, roundtables, and conferences are several examples | Experts: the IPCC authors were usually the ones giving presentations, answering questions etc. | Very diverse, depending on the event. Audiences include researchers, policymakers, and businesspeople. |
| 5 | We explore possibilities to empower people to reflect on their eating preferences by organizing protein demonstrations for Finnish students aged 10–16 | A demo on new sources of protein (meat alternatives) to encourage sustainable eating. Students got to taste and discuss several alternatives to meat. | Expert: The researchers designed and executed the protein demonstrations. While the students were tasting the foods, the researchers offered them information about their production, use, environmental impacts, and nutritional value. | 230 students from two secondary schools. Demos were done in groups ranging from 3-20 participants. |

| | | | |
|---|---|---|---|
| 6 | The goal of this work is to explore how stakeholders' intention of engaging in adaptation to climate change can be explained (Study 1) and increased (Study 2). NOTE: For the review we merely looked at results from study 2, since this involved the evaluation of a climate communication activity | Two local workshops on adaptation to climate change.<br><br>The first workshop's goal was to gather stakeholder ideas for local adaptation measures, whereas the second workshop's goal was to discuss opportunities and constraints for the implementation of the most promising measures. | Expert: The scientists elaborated on the ideas proposed in workshop 1 and summarized the adaptation measures for workshop 2. It is unclear whether the scientists also interacted with the stakeholders during the workshops | Stakeholders such as policymakers, government departments and administration (local, regional, national), non-governmental organizations with environmental, economic, and social interests (local, regional, national), local business and industry, local communities, and researchers working on climate change issues (regional and national) |
| 7 | This paper describes how an Australian politician in a position of power changed his mind about climate change, in response to deliberations of a panel of scientists broadcast on television. | Broadcast on climate change by scientists on television | Expert: Five scientists who participated in a panel discussion on climate change, during a 'science special' TV broadcast | A climate sceptic politician |

Table 2:  Overview of the studies included in the analysis.

## 4 Results

### 4.1 Role of the scientist

In four studies (Calyx & Low, 2020; Jacobson et al., 2016; Pathak et al., 2021; Peltola et al., 2020) the role of the scientists was mostly to act as a source of information for the target audience. For example, by answering questions (Jacobson et al., 2016) or providing information on climate change to the public (Calyx & Low, 2020; Pathak et al., 2021; Peltola et al., 2020). In one study, scientists were involved in designing and building an exhibition (Kluczkovski et al., 2020) but not in the actual exhibition itself. In another study (Luís et al., 2018), it is unclear to what extent scientists interacted with the public but here scientists were involved in designing workshops on adaptation strategies on climate change for the target audience. We see these roles as examples of deficit communication where scientists are mainly involved as experts.

In only one study, scientists were really collaborating with members of the public by writing poetry on climate change and engaging in discussion with participants during workshops (Illingworth & Jack, 2018). Here, the scientists aimed to get a better understanding of the questions and feelings of these participants about climate change and to create a dialogue between scientists and non-scientists. Interestingly, the authors refer to the involved scientists as 'experts'. According to Metcalfe (2019), these are examples of a dialogue approach. Furthermore, this is the only study where the participating

scientists were asked to reflect on their interaction with the public. None of the other studies describe how scientists experienced the communication activity.

## 4.2 Output: reached audiences

In terms of *output*, the audiences that were reached in the different studies are quite diverse both in number and in characteristics. For example, Peltola et al., (2020) were able to reach 230 secondary school students, whereas Illingworth and Jack (2018) specifically aimed to reach people from communities who are usually underserved by science communication, such as refugees. They were able to reach approximately 21-31 people from these communities. On the other hand, Calyx and Low (2020) describe the impact of a TV broadcast where scientists explained about climate change on a climate sceptic politician. It is encouraging to see that climate scientists aim to reach diverse audiences, because that might mean that different groups within society have access to climate communication activities.

## 4.3 Outcomes and impact of climate communication activities

All the included studies show positive results regarding *outcomes*. In three of the seven included studies, a pre- posttest design was used to measure for example knowledge about climate change (Jacobson et al., 2016; Kluczkovski et al., 2020), behavioral intention to engage with climate change (Luís et al., 2018) and actual behavior change (Kluczkovski et al., 2020). In these studies, significant increases in the measured variables were found. In three other studies, the authors conclude that their activities provided a platform for sharing views, experiences, and create open discussions about (topics related to) climate change (Illingworth & Jack, 2018; Pathak et al., 2021; Peltola et al., 2020). Calyx and Low (2020) found that a TV broadcast where a panel of scientists discuss climate change made a climate sceptic politician change his mind about climate change.

None of the included studies evaluate the outcomes over a longer period of time, or changes in for example social norm of a specific societal group. We therefore conclude that in none of the studies the *impact* of the climate communication activities is evaluated.

## 5 Discussion

The goal of this study was to gain insight into what is known about the impact of climate communication activities by scientists and how scientists can make an impact when partaking in climate communication activities for different audiences. However, our most important finding is that there is hardly any research that evaluates the impact of climate communication activities by scientists. This does not necessarily mean that communication activities aren't evaluated. Ziegler et al., (2021) suggest that evaluations of science communication (in general) are often used to reflect upon a project or activity within a team to improve future projects and the results are often only shared with supervisors or funders. Ziegler et al., (2021) also

found that many of evaluation reports are used as summative evaluations to determine the 'successes' of a project rather than formative evaluations that could help gain a deeper understanding of the development and process of activities.

Of course, we might have missed evaluations because of too stringent search terms (Table 1). We could have missed relevant keywords, although for example a test to extend the search with the keywords 'dialogue' and 'participatory' only seemed to lead to false positives. Alternatively, evaluations of science communication activities could have been published outside of the peer-reviewed literature, as for example internal reports, blog posts or conference presentations.

A possible reason for not making evaluation outcomes available, is that negative results of evaluations might lead to
criticism. This could be especially problematic if this impacts funding opportunities or the way scientists who are involved in such activities are seen by peers and the wider public. However, sharing evaluations of climate communications efforts both successful and unsuccessful could stimulate learning and help other scientists get a better understanding of how for example, specific groups of people could be approached, or what activities (e.g., lectures, hands-on activities, group discussions) might be suitable for achieving specific goals (e.g., improving knowledge, influencing attitude etc.).

The seven included peer-reviewed studies showed positive results regarding the outcomes of climate communication activities by scientists. This is encouraging, for this seems to indicate that climate communication activities by scientists are valuable in providing platforms for discussion, reflection and could stimulate the public to become more engaged with climate change. However, care should be taken in concluding from these seven articles that climate communication activities by scientists always have a positive impact; especially considering the suggestion by Ziegler et al., (2021) that positive
results are more likely to be shared than negative results. One such an example is the study by Calyx and Low (2020) which was included in this review. This study describes the positive impact of a TV broadcast where a panel of scientists discuss climate change made a climate sceptic politician change his mind about climate change. Calyx and Low state that they wanted to "share a story of change" (p. 3) which made them decide to, among other things, write a paper about the impact of the TV broadcast. It is possible that the paper had not been written, if there had been no impact. We thus encourage the
publication of evaluations where the effect of the communication activity is non-positive, too.

The use of existing theoretical frameworks or models from fields such as education and communication can help guide the design and evaluation of climate communication activities. One of the included studies (Luís et al., 2018) is an example of this. In this study the Theory of Planned Behavior (Ajzen, 1991; 2001) was used. Using this theory, Luís et al., (2018) selected variables (attitude, subjective norm, and perceived behavioral control) that might impact stakeholders' intention to
plan local adaptations to climate change. Then, they determined how these variables could be defined and measured and evaluated whether these variables were impacted by a climate communication activity (in this case two local workshops on adaptation to climate change).Another finding of this review is that in six of the seven studies, scientists take on a role of expert and engage in one-way communication. Either by being a 'source of information' (answering questions, but not asking about the views or ideas of the public) or by being involved in designing climate communication activities but not
being involved in actual interaction with the public. These are examples of a deficit approach to climate communication. This is finding is similar to the findings of Metcalfe (2019) who found that most science communication activities (in

Australia) use a deficit approach, despite the often called-for shift in many countries from 'communication' to 'dialogue' and from 'science and society' to 'science in society' (Bucchi, 2008). There has been a lot of critique on the deficit approach to science communication. The most important being that using a deficit approach assumes that providing the public with information about a scientific issue will 'correct' or 'complement' their views, making sure people 'believe the right things' (Seethaler et al., 2019). However, human reasoning is complex and attempts to correct 'misconceptions' by merely providing information often backfires (Lewandowsky et al., 2012). A possible reason for the consistent use of a deficit approach could be that scientists are hardly ever trained in science communication (Simis et al., 2016). Therefore, scientists might not be aware of the limitations of using a deficit approach. In our review, we found that in only one study (Illingworth & Jack, 2018) scientists were asked to reflect upon their interaction with the public. However, reflecting on communication experiences could provide insight into the perspectives of the scientists on science communication and whether and which support they might need to become better communicators. We thus call for more interdisciplinary research where scientists and communication professionals collaboratively investigate the impact of communication activities, specifically those using dialogue and participatory models.

Fortunately, evaluation has become a much more important topic in science communication research in the very recent past (e.g., Hillier et al. 2021). Between the finalisation of our analysis and the revision of our manuscript, new, relevant manuscripts have come to our attention (e.g., Hiller and Van Meeteren, 2024). While we decided not to open up our analysis again (because a manuscript like this one would then never be finished), we do expect that an update of this search could be very valuable in a few years.

Future research could focus on the role of non-scientists, such as knowledge brokers (Meyer, 2010) and other communicators, in climate science communication. Such future research could also extend our analysis by focussing on the effectiveness of climate scientists in communicating or training to professionals, rather than the general public.

## 6 Conclusions

Based on the review presented in this paper, our most important conclusion is that more research is needed on evaluating the impact of climate communication activities by scientists. To achieve this, we believe that a learning-friendly environment is necessary, where the focus of studies is not necessarily on 'proving successes' but on sharing lessons learned from evaluating climate communication activities with the goal of increasing understanding of what works and what not.

Furthermore, collaboration between researchers and science communicators could help in setting up more systematic approaches to evaluating (science) communication activities (Ziegler et al., 2021). The experiences of science communicators can help researchers get a better idea of the challenges that are faced when evaluating climate communication activities and the expertise of social scientists might help to overcome such challenges. This collaboration could help in setting up evaluation practices that are methodologically sound and fit the goals of the climate communication activity.

One challenge in evaluating science communication (in general) is the method of evaluation. There are several papers that
describe approaches and methods that can be used to evaluate science communication activities (e.g., Grand & Sardo, 2017; Peeters et al., 2022; Ziegler et al., 2021) that might be useful to consider when evaluating climate communication activities by scientists.

We recommend that evaluation of climate communication does not only focus on the impact of these activities on the public, but also on the experiences of the involved scientists. These experiences could provide valuable insights into whether the involved scientists need support or training (Baram-Tsabari & Lewenstein, 2017) to help them become better climate communicators.

## Acknowledgements

We thank the two reviewers (Usha Harris and Angelica Alberti-Dufort) for their kind and constructive feedback, which have helped improve our manuscript.

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

**\*Studies included in the review.**

**Appendix A**


List of studies included in this review:

Calyx, C., & Low, J. (2020). How a climate change sceptic politician changed their mind. *Journal of Science Communication, 19*(3), 1-12. https://doi.org/10.22323/2.19030304

Illingworth, S., & Jack, K. (2018). Rhyme and reason-using poetry to talk to underserved audiences about environmental

change. *Climate Risk Management, 19,* 120-129. https://doi.org/10.1016/j.crm.2018.01.001

Jacobson, S., Seavey, J., & Mueller, R. (2016). Integrated science and art education for creative climate change communication. Ecology and Society 21(3), 1-6. http://dx.doi.org/10.5751/ES-08626-210330

Kluczkovski, A., Cook, J., Downie, H., Fletcher, A., McLoughlin, L., Markwick, A., Bridle, S., Reynolds, C., Schmidt Rivera, X., Martindale, W., Frankowska, A., Moraes, M., Birkett, A., Summerton, S., Green, R., Fennel, J., Smith, P.,

Ingram, J., Langley, I., … & Ajagun-Brauns, J. (2020). Interacting with members of the public to discuss the impact of food choices on climate change-experiences from two UK public engagement events. *Sustainability, 12,* 1-21. https://doi.org/10.3390/su12062323

Luís, S., Lima, M., Roseta-Palma, C., Rodrigues, N., Sousa, L., Freitas, F., Alves, F., Lillebø, A., Parrod, C., Jolivet, V., Paramana, T., Alexandrakis, G., & Poulos, S. (2018). Psychosocial drivers for change: Understanding and promoting

stakeholder engagement in local adaptation to climate change in three European Mediterranean case studies. *Journal of Environmental Management, 223,* 165-174. https://doi.org/10.1016/j.jenvman.2018.06.020

Pathak, M., Roy, J., Patel, S., Some, S., Vyas, P., Das, N., & Shukla, P. (2021). Communicating climate change findings from IPCC reports: insights from outreach events in India. *Climatic Change, 168*(23), 1-14. https://doi.org/10.1007/s10584-021-03224-8

Peltola, T., Kaljonen, M., & Kettunen, M. (2020). Embodied public experiments on sustainable eating: demonstrating alternative proteins in Finnish schools. *Sustainability: Science, Practice and Policy, 16*(1), 184-196. https://doi.org/10.1080/15487733.2020.1789268