# Peer review of "Evaluating the impact of climate communication activities by scientists: What is known and necessary?"

_EGUsphere, 2023_

## Community Comment (CC1)

**Caveat:** This is a brief community comment, in which the authors and editors should note a potential conflict of interest as I am an Exec. Editor of Geoscience Communication [Not handling editor for this manuscript] and because I'm clearly pointing to my own work.

**Comment:** I do not pretend a full overview of this subject area, but have a few specific pieces of knowledge that the authors might consider incorporating.  The two non-editorials might either be outside the scope of the authors definitions of 'communicating climate science', or out of scope due to their publication/submission date. However, very briefly noting them might help (i) sharpen their definition/scope which I didn't read as excluding the suggestions I make and (ii) and potentially recognise upcoming work.

**Use the community?** There may also be an opportunity to ask the community to contribute examples they know by commenting on the discussion paper.

**Why are the following out of scope?**
- https://gc.copernicus.org/articles/1/35/2018/ - Contains a communication activity that is evaluated.
- Suggest the authors review papers published in *Geoscience Communication*.

**Detail of 4 papers:** In Section 2.1 'Importance of Evaluation', it might be worth nothing that this aligns entirely with *GC*s principles for publication as outlined in two editorials
- https://gc.copernicus.org/articles/1/1/2018/
- https://gc.copernicus.org/articles/4/493/2021/

There is also a paper of mine that was in *GC Discussions* since June 2023 on '*A tool to co-create impactful university-industry projects for natural hazard risk mitigation'*, which contains an evaluation (Case Study) of a project on climate-driven hydrological risk. https://egusphere.copernicus.org/preprints/2023/egusphere-2023-1251/
The project output is a blog published by the Bank of England - https://bankunderground.co.uk/2021/04/08/its-windy-when-its-wet-why-uk-insurers-may-need-to-reassess-their-modelling-assumptions/

I have also just submitted a GC Insights paper (after the submission of your manuscript) explicitly to evaluate a project to communicate climate related risk.
- The output of that work is https://bankunderground.co.uk/2023/04/13/what-if-its-a-perfect-storm-stronger-evidence-that-insurers-should-account-for-co-occurring-weather-hazards/
- 'Open R-code to communicate the impact of co-occurring natural hazards' - egusphere-2023-2799. https://egusphere.copernicus.org/preprints/2023/egusphere-2023-2799/

All the best,

John Hillier

---

## Author Comment (AC1)

We thank John Hillier for his insightful community comment. Below, we briefly respond (in black) to his comments (in blue):

**Caveat:** This is a brief community comment, in which the authors and editors should note a potential conflict of interest as I am an Exec. Editor of Geoscience Communication [Not handling editor for this manuscript] and because I'm clearly pointing to my own work.

**Comment:** I do not pretend a full overview of this subject area, but have a few specific pieces of knowledge that the authors might consider incorporating. The two non-editorials might either be outside the scope of the authors definitions of 'communicating climate science', or out of scope due to their publication/submission date. However, very briefly noting them might help (i) sharpen their definition/scope which I didn't read as excluding the suggestions I make and (ii) and potentially recognise upcoming work.

**Use the community?** There may also be an opportunity to ask the community to contribute examples they know by commenting on the discussion paper.

We absolutely agree that our manuscript is an opportunity to contribute examples by the community, which is one of the reasons why we submitted our manuscript to *Geoscience Communication*. This specific Community Comment is a good example of such a discussion, fully in the spirit of Open Science.

**Why are the following out of scope?**

- https://gc.copernicus.org/articles/1/35/2018  - Contains a communication activity that is evaluated.

This is indeed an interesting manuscript. We have looked through our data, and this manuscript was not picked up in our literature search because it did not contain all the search terms (Table 1) in the title, abstract or keywords. In particular, "Climate change" does not appear in the title, abstract or keywords of https://gc.copernicus.org/articles/1/35/2018. We prefer to keep our original literature search strategy, which was carefully crafted to be a balance between breadth and our specific scope to climate change issues (rather than all geoscience).

- Suggest the authors review papers published in *Geoscience Communication*.

Our initial literature search (Table 1) did pick up an article in *Geoscience Communication* (https://gc.copernicus.org/articles/3/381/2020/), but that did not pass through our subsequent selection criteria.

**Detail of 4 papers:** In Section 2.1 'Importance of Evaluation', it might be worth nothing that this aligns entirely with *GC*s principles for publication as outlined in two editorials

- https://gc.copernicus.org/articles/1/1/2018
- https://gc.copernicus.org/articles/4/493/2021

This is a good idea; we will refer to one or both of these editorials in our revised manuscript.

There is also a paper of mine that was in *GC Discussions* since June 2023 on '*A tool to co-create impactful university-industry projects for natural hazard risk mitigation'*, which contains an evaluation (Case Study) of a project on climate-driven hydrological risk. https://egusphere.copernicus.org/preprints/2023/egusphere-2023-1251

While an interesting article, we prefer to keep our literature search to articles published between 2012 and 2022. A paper like this will otherwise never be finished. Nevertheless, highlighting these newer preprints in this comment is very valuable. We therefore propose to add a paragraph in the Discussion of the revised manuscript where we highlight that new papers have come to our attention in the review phase.

The project output is a blog published by the Bank of England - https://bankunderground.co.uk/2021/04/08/its-windy-when-its-wet-why-uk-insurers-may-need-to-reassess-their-modelling-assumptions/

I have also just submitted a GC Insights paper (after the submission of your manuscript) explicitly to evaluate a project to communicate climate related risk.

- The output of that work is https://bankunderground.co.uk/2023/04/13/what-if-its-a-perfect-storm-stronger-evidence-that-insurers-should-account-for-co-occurring-weather-hazards/
- 'Open R-code to communicate the impact of co-occurring natural hazards' - egusphere-2023-2799. https://egusphere.copernicus.org/preprints/2023/egusphere-2023-2799

Again, we prefer to keep our literature search to articles published between 2012 and 2022.

---

## Author Comment (AC2)

We thank Angelica Alberti-Dufort for her insightful review. Below, we respond (in black) to the comments (in blue):

I will start by saying that this review was very insightful. I'm both surprised and disappointed by the lack of literature regarding this subject and I agree that this should be better understood and studied, as I work everyday with climate scientists to help them better communicate.

We thank the reviewer for this supportive comment on our manuscript.

Here are some comments. I hope this can help you improve your paper.

I suggest that you dig a little deeper into the literature about scientists as communicators in general, not just in the field of climate science. I know that there is a lot of literature that delves into this more general issue.

The reviewer is right that there is a very extensive literature on scientists as communicators, including a suite of academic journals and dedicated toolboxes focussing on the topic. We feel that an overview of the general topic would never be exhaustive and would thus not do the field justice. We thus prefer to focus on climate scientists as communicators, also because the challenges in communicating about climate science are different from many other fields of science. Climate change is abstract and emotive (scary) for many people, so in this study we want to focus on effective ways in which climate scientists can communicate about this topic.

For instance, in Section 1.1, it is mentioned that research shows that scientists are seen by the public as trusted information producers and that they should increase their communication efforts, which is true, but a limited description of scientists as communicators. This positive aspect about scientists can sometimes be counterbalanced by recurring shortcomings such as the difficulty in simplifying scientific information or the sense of inferiority that non-scientific target audiences may feel towards them. Additionally, scientists generally lack the necessary skills and tools to communicate effectively based on their target audience, just because they weren't trained to do so.

This is a good point by the reviewer. In the revised manuscript, we will add a sentence that insufficient training makes scientists hesitant to engage in climate communication activities, and refer to Rozance et al (2020; https://dx.doi.org/10.1088/1748-9326/abc27a).

These observations about scientists, in general, could also be discussed in section 5, where it is mentioned that care should be taken in concluding that climate communication activities by scientists have a positive impact.

The reviewer is right that communication activities by scientists might not always have a positive impact. All studies in our sample did report a positive impact, but this may be due to publication bias, where only 'positive' results tend to be published. One of the key messages of our manuscript is that also non-positive impacts should be evaluated and published. We will further clarify this point in the revised manuscript.

The emergence of "knowledge brokers" in the last decade is something worth exploring to enhance scientific communication. These brokers are particularly present in various health science fields. They may or may not be scientists, but they possess the technical skills to communicate science and engage their audience while maintaining scientific rigor. They are an important tool to scientists who need to communicate. https://journals.sagepub.com/doi/abs/10.1177/1075547009359797

We thank the reviewer for this comment; but feel that knowledge brokers are a bit out of scope for this review. We would like to keep the focus on the climate scientists as communicators. The role of knowledge brokers could very well be the topic of another review analysis, and we will in the revised manuscript add it to a new paragraph in the discussion section about possible future research directions.

This brings me to suggest to add some words about who are the main climate communicators. Except for scientists, who else is talking about climate change and what can we say about them (these include knowledge brokers).

Again, we prefer to keep the focus of our manuscript on the scientists, as this was the original aim of our research. We want to avoid 'research creep', by extending the aims beyond what we set initially. While it would be very interesting to see an overview of all players in the climate communication space, that would be beyond this manuscript. We will thus mention this in the new paragraph in the discussion section about possible future research directions.

Along these lines, in Section 2.2 : You mention "scientists" and "science communicators" which for me, are very different source of information with different sets of skills and different impacts on the target publics. I get that these two types are described in the dialogue model, but when you read the paper it feels like these two are the same. If you have better presented the different communicators somewhere between the introduction and the theoretical framework, it will be less confusing.

We understand that this indeed was confusing in the original version of the manuscript. In the revised version, we will remove 'science communicators' from this paragraph on the dialogue model and only refer to scientists; also to emphasize the scope of our article.

I feel like the fundamental goal of the review, or what the conclusions will lead to, could be more specific. For instance, if there are so few research papers on this subject what should we do about it? If the deeper goal is to have effective communications that reach as many people as possible and have the greatest impact on their engagement in the climate change crisis, regardless of the communicator's status, I believe one of the main conclusions should be the need to explore the possibilities offered by knowledge broker or other types of communicators in collaborating with climate scientists.

We do not entirely agree with the reviewer's suggestion that the goal of our manuscript is to have effective communications that reach as many people as possible and have the greatest impact on their engagement in the climate change crisis. Instead, the goal is to improve the quality of the science communication by scientists, by evaluating the impact of

communication activities. As we also responded above, we prefer to leave knowledge brokers to a dedicated article.

Helping climate scientists become better communicators is also a worthwhile goal but very different to me. Both could be discussed in the paper. https://link.springer.com/chapter/10.1007/978-3-319-50398-1_22

We thank the reviewer for this reference; and agree that it is a useful addition. We will add it to the conclusion section of the revised version of our manuscript.

Training for professionals, outside the academic setting, is an important part of climate change communication activities, but there is little evidence of its effectiveness in changing perceptions and behavior. This type of communication is perhaps more often associated with scientists and could be interesting to explore in future research. « Il n'existe aucune preuve de l'impact d'une formation de sensibilisation aux enjeux climatiques sur les comportements » (lemonde.fr)

We feel that teaching for professionals is a bit outside the scope of our research, so prefer not to explicitly discuss it in this manuscript. It might be relevant for a follow-up study, though. We will add it to the new paragraph in the discussion section about possible future research directions.

Section 4.2: The audiences accessing climate information could be something better documented in other types of papers who might have been excluded from your review because of your methodological choices. I'd end this section by opening on what we could find elsewhere on this matter, like the fact that these conclusions can vary greatly across the world. For example, here in Canada, scientists and climate communicators are aiming to a very large public as a lot of them are government employees who have the "mandate" to inform practitioners and the population in general, about climate mitigation and adaptation. But this couldn't be more false in other countries.

We don't intent to provide an exhaustive list of audiences reached by *all* climate science communication in section 4.2, which is about the results of our analysis. Instead, we aim to give an overview of the audiences reached by those seven articles that pass our selection criteria. We thus think that including a comment here about other audiences might confuse readers.

Finally, it might be interesting to add a few words about the different target audiences for climate change-related information and their levels of knowledge and engagement. This would help support the argument presented in section 1, line 25. https://www.nber.org/system/files/working_papers/w30265/w30265.pdf

We thank the reviewer for this useful reference, and will add it to the introduction of our revised manuscript.

In the conclusions, you talk a lot about communication frameworks (which is ok, because it is what you chose to research about) but you could also suggest more research on tools that could help climate scientists to communicate. One important tool is to clearly define the target

audience. For example, this survey is prepared each year by a marketing communication research lab in Laval university in Quebec, Canada. It is a tool to help Quebec's climate communicators get to know their audience better and prepare their interventions (french paper). https://unpointcinq.ca/wp-content/uploads/2023/11/Barometre-Action-Climatique-2023.pdf

To keep the focus of our manuscript clear, we would prefer not to digress into a discussion of tools. That would warrant its own review (which would be very useful to the field!) but we wouldn't be able to do that justice within the scope of this manuscript.

---

## Author Comment (AC3)

We thank Usha Harris for her insightful review. Below, we respond (in black) to the comments (in blue):

This manuscript reports on original research conducted by the authors. The aim, study design and results are communicated with excellent clarity. It makes a much-needed contribution to the field of science communication by evaluating the impact of climate communication activities by scientists on ordinary people's behaviour of which there is limited knowledge, as the study finds.

We thank the reviewer for these extremely kind words; they are much appreciated.

The title is clear and aptly describes the content of the manuscript. The abstract provides a short and clear summary of the important findings and conclusions? The introduction provides a good summary of literature on the topic with well-defined aim and research question. The framework for evaluating science communication activities is identified along with the three communication models used by scientists. The authors may like to include examples of when each of these models may have been used effectively. For example, there is a place for the deficit model in disaster risk communication in the aftermath of disasters, but dialogue and participatory approaches may be more effective in designing effective risk communication strategies with communities.

We thank the reviewer for this suggestion. We will indeed add a comment in the revised manuscript about when these roles are appropriate.

For the deficit model: *"This may sometimes be an appropriate role, for example in disaster risk communication."*

For the dialogue model: *"This is an appropriate role when for example designing effective risk communication strategies with communities and could also help the scientists to improve their research (Boon et al., 2022)."*

The method for literature survey is well explained. Did the researchers consider using participatory and dialogue as climate change communication search terms since these methods are considered to be more effective in bringing about behaviour change?

The reviewer addresses a good point. We did include citizen engament and public involvement as search terms for the climate communication aspect, but not participation or dialogue. We ran our search again including the terms "dialogue" and "participatory" and evaluated the impact of including these terms. We found that the inclusion of these terms mostly brought up false positives about engagement activities without scientist involvement. We therefore decided not to redo our literature search.

I would suggest that authors include the description of the included studies as a table within the text instead of as Appendix A. The discussion highlights some important findings including the need to include both negative and positive impacts of communication activities.

This is a good suggestion. In the revised manuscript, we will move the table from the Appendix into the main text (as a new Table 2).

*The authors may like to suggest the importance of interdisciplinary studies where scientists and communication researchers can work collaboratively to understand impact of communication activities especially using dialogue and participatory models.*

This is a good point. We will indeed add such a suggestion to the end of the discussion section:

*"We thus call for more interdisciplinary research where scientists and communication professionals collaboratively investigate the impact of communication activities, specifically those using dialogue and participatory models."*

*Overall, this manuscript is very well written and offers an important insight into an area which is in need of further exploration.*

We thank the reviewer for this very kind and supportive feedback

---

## Author Response (AR1)

Dear Louise, thank you for your comments on our responses to the reviews. Below, we briefly respond (in black) to your comments (in blue):

Thank you for engaging so thoroughly in the discussion process. As per your responses to the three comments (CC1, RC1 and RC2), please incorporate the changes you propose into your revised manuscript.

Yes, we will do. We already have a revised version of the manuscript ready, but are waiting for that step in the GC process.

In addition, please consider these additional comments from myself:

- As per RC2, please consider briefly highlighting the different actors of climate communication (e.g., knowledge brokers) in the introduction (i.e., not only discussion). While your focus on scientists, partly to keep the scope of your article realistic, makes sense, I believe that briefly mentioning other actors that exist in this sphere would help set the scene a bit more completely and do these various actors' and their important work justice.

This is a good idea. We have now added a sentence to the first sentence of section 1.1. in our revised manuscript: "While there are other actors who engage in climate communication (e.g. communication professionals, knowledge brokers), we chose to […]"

- Please state that you use the different roles of scientists for your literature evaluation (section 2.2), as you did for the three levels of evaluation at the end of section 2.1.

A good point. We have now added a section to the end of section 2.2: "Again, we used these three roles of scientists to review the climate communication activities that are described in the literature."

- In your process for the literature review, you mention selecting articles based on whether they describe the "impact" of climate communication activities. Yet in section 2.1 you highlight two lower levels ("output" and "outcomes"), which you report on in the results. And none of the articles you retained evaluate the "impact" the climate communications activities, as reported in section 4.3. Can you please address this inconsistency between your methods and the results?

This is a very good comment; you are right that this was an inconsistency in the original manuscript. In the revised manuscript, we now rephrase the sentence in section 3.1 to "(2) the study describes *the output, outcome, and/or impact* of climate communication activities."

- Please consider adding key outputs/outcomes/impacts (point 3 of your analysis, sections 4.2 and 4.3) to your table in Appendix A.

We have tried this in an earlier version of the manuscript, but the table became too unwieldy; as it was sometimes difficult to summarise the output, outcome and/or impact in a few sentences. We thus prefer to keep it as is.

- From your evaluation of these 7 articles, can you comment on whether there are any patterns emerging on the type of activity, the audience reached, and the goal of the study? For example, are there specific activity formats that seem to be preferentially used with specific audiences or goals (and ultimately impacts)? This article may be of interest: https://doi.org/10.5194/gc-2-39-2019

We have looked into whether we could discern any patterns, but seven articles are just too few to make meaningful/robust statements. We don't want to overspeculate and therefore leave the analysis of patterns to a potential follow-up study in a few years' time, when hopefully the number of articles on the evaluation of science communication activities by climate scientists has increased significantly.

 - You could consider writing "outputs" and "outcomes" in italics in sections 4.2 and 4.3, like you do for "impacts", so we quickly make the link with the three levels of evaluation you present in section 2.1.

This is a good idea; we have changed this in the revised manuscript.

- Please address potential limitations of your search criteria in a bit more depth in the discussion. Here are some ideas of topics that could be addressed: potentially missed articles due to the selection of keywords, outcomes not published as articles, but in other formats such as internal reports (which you mention), blog posts, or conference presentations.

We have added a paragraph to the discussion section of the revised manuscript:

"*Of course, we might have missed evaluations because of too stringent search terms (Table 1). We could have missed relevant keywords, although for example a test to extend the search with the keywords 'dialogue' and 'participatory' only seemed to lead to false positives. Alternatively, evaluations of science communication activities could have been published outside of the peer-reviewed literature, as for example internal reports, blog posts or conference presentations.*"

- This GC editorial currently in discussion highlights support for scientists to become better communicators, among other things, and could be of interest: https://doi.org/10.5194/egusphere-2023-3121

While this is a very interesting editorial, we prefer to keep our cited literature to published papers as much as we can. We are confident that your editorial will get the attention and recognition that it deserves as soon as it is published.

- It is not very common to include new content/references in the conclusion section (one-to-last paragraph). Please consider moving this new content earlier in the paper instead, perhaps in the discussion section.

We have moved this paragraph to the discussion section in the revised version of the manuscript.

- I am not sure whether your approach for highlighting studies included in the review (*) will work with the format of GC papers. If it doesn't, these could perhaps be identified in a new table in the appendix instead, as I think it's indeed useful to highlight them.

Yes, this is a good idea. We have included a list of the seven articles in Appendix A in the revised manuscript.